# Characterization of Two Novel Rumen-Derived Exo-Polygalacturonases: Catalysis and Molecular Simulations

**DOI:** 10.3390/microorganisms11030760

**Published:** 2023-03-16

**Authors:** Qian Deng, Xiaobao Sun, Deying Gao, Yuting Wang, Yu Liu, Nuo Li, Zhengguang Wang, Mingqi Liu, Jiakun Wang, Qian Wang

**Affiliations:** 1Key Laboratory of Molecular Animal Nutrition, Ministry of Education, Zhejiang University, Hangzhou 310058, China; 2Institute of Dairy Science, College of Animal Sciences, Zhejiang University, Hangzhou 310058, China; 3Key Laboratory of Marine Food Quality and Hazard Controlling Technology of Zhejiang Province, College of Life Sciences, China Jiliang University, Hangzhou 310018, China; 4College of Life Sciences, Zhejiang University, Hangzhou 310058, China

**Keywords:** rumen microbe, exo-polygalacturonase, expression, hydrolytic pattern, molecular dynamics stimulation

## Abstract

Pectinases are a series of enzymes that degrade pectin and have been used extensively in the food, feed, and textile industries. The ruminant animal microbiome is an excellent source for mining novel pectinases. Two polygalacturonase genes, *IDSPga28-4* and *IDSPga28-16*, from rumen fluid cDNA, were cloned and heterologously expressed. Recombinant IDSPGA28-4 and IDSPGA28-16 were stable from pH 4.0 to 6.0, with activities of 31.2 ± 1.5 and 330.4 ± 12.4 U/mg, respectively, against polygalacturonic acid. Hydrolysis product analysis and molecular dynamics simulation revealed that IDSPGA28-4 was a typical processive exo-polygalacturonase and cleaved galacturonic acid monomers from polygalacturonic acid. IDSPGA28-16 cleaved galacturonic acid only from substrates with a degree of polymerization greater than two, suggesting a unique mode of action. IDSPGA28-4 increased the light transmittance of grape juice from 1.6 to 36.3%, and IDSPGA28-16 increased the light transmittance of apple juice from 1.9 to 60.6%, indicating potential application in the beverage industry, particularly for fruit juice clarification.

## 1. Introduction

The plant structural polymers cellulose, hemicellulose, pectin, and lignin are the most abundant renewable organic carbon source in nature. Pectin, the most complex of these polysaccharides, gives plants structural rigidity and resistance to attack by insects and pathogens [1,2]. Pectin mainly comprises homogalacturonan regions (also known as “smooth” regions) consisting of D-galacturonic acid units, linked by α-1,4 glycosidic bonds, and substituted homogalacturonan regions (also known as “hairy” regions) such as xylogalacturonan, rhamnogalacturonan I, and rhamnogalacturonan II, which account for 15–30% of the pectin in fruits and vegetables [3,4]. Complete hydrolytic degradation of pectin requires a series of functional pectic enzymes, because of its complexity and heterogeneity. These enzymes include polygalacturonase (three types), pectate lyases (EC 4.2.2.2 and EC 4.2.2.9), pectin methylesterase (EC 3.2.1.11), and rhamnogalacturonase (EC 3.2.1.171) [5,6]. Breakdown of homogalacturonan, the simplest structural region in pectin, is initiated by endo-polygalacturonase (EC 3.2.1.15), which attacks the α-1,4 glycosidic bonds between D-polygalacturonic acid residues and generates pectic oligogalacturonides. Exo-polygalacturonase (EC 3.2.1.67) or exo-poly-α-digalacturonosidase (EC 3.2.1.82) then cleave the first glycosidic bond from the non-reducing end of the oligogalacturonides to release monomers or dimers.

Pectinases have been used extensively in the food, feed, and textile industries, because of their effectiveness in degrading plant material, such as fruit pulp. According to the carbohydrate-active enzyme (CAZyme) database (http://www.cazy.org/, accessed on 10 March 2022) [7], these pectinolytic enzymes are mainly classified into the glycoside hydrolase (GH) family 28. Many GHs have been identified from bacteria, fungi, and higher plants and more than 12,000 GH28 gene sequences have been deposited in GenBank, but only about 100 of them have been functionally characterized. For instance, NfPG I, derived from *Neosartorya fischeri* P1, was reported as the most catalytically active endo-polygalacturonase with a specific activity of 40,123 U/mg [8]. To satisfy the demands of industrial bioprocessing, a highly active (25,900 U/mg) endo-polygalacturonase, TePG28b, with good thermostability (optimum temperature 70 °C), was isolated from a thermophilic fungus, *Talaromyces leycettanus* [4]. The hydrolytic behavior, catalytic mechanism, and molecular basis of endo-acting polygalacturonases have also been well-elucidated [9,10,11]. In contrast, to our knowledge, characterized exo-polygalacturonases and exo-poly-α-digalacturonosidases are much less catalytically efficient, with specific activities <300 U/mg [5,12,13,14].

Ruminants, such as sheep and cattle, digest nutrient-poor plant material, such as grass, very efficiently, because of the numerous cellulolytic bacteria and fungi in the rumen. A variety of CAZymes including glucanase [15,16], cellulase [17], and xylanase [18] have been mined using multi-omics sequencing and heterologous expression strategies. Apart from degradation by CAZymes, the metabolism of complex polysaccharides such as pectin is also orchestrated by gene clusters, termed polysaccharide utilization loci (PULs), found in gut microbes, particularly in the Bacteroides genus [19,20]. Thus, gastrointestinal microbes in ruminants are of tremendous interest for mining novel CAZymes for industrial application. In our previous study, ~15,000 unigenes including 4225 open reading frames (ORFs) were obtained from the sheep rumen microbiome using a metatranscriptomics strategy [17]. Of these, 173 ORFs were annotated to the GH28 family. In order to explore highly efficient pectinases, we randomly picked ten of the 173 ORFs for gene cloning and succeeded in obtaining two full-length GH28 family proteins. In this study, the two GH28 polygalacturonase genes, *IDSPga28-4* (GenBank: OK149199) and *IDSPga28-16* (GenBank: OK149200), were cloned from rumen fluid cDNA and heterologously expressed in *Escherichia coli*. The biochemical properties, hydrolysis products, and molecular basis for catalysis of the recombinant enzymes were investigated.

## 2. Materials and Methods

### 2.1. Materials

Sheep rumen cDNA was prepared as described previously [17], stored at −80 °C, and used within one month. The plasmid pET-30a (+) was from Invitrogen (Shanghai, China), and expression host *E. coli* BL21 (DE3) was from Tiangen (Beijing, China). Polygalacturonic acid (PGA) from citrus pectin and rhamnogalacturonan I from potato were from Megazyme (Wicklow, Ireland).

D-(+)-Galacturonic acid (GalA) monohydrate, digalacturonic acid ((GalA)_2_), trigalacturonic acid ((GalA)_3_), and pectin from citrus peel (~60% degree of esterification) were from Sigma-Aldrich (St. Louis, MO, USA). Isopropyl-thio-β-D-galactopyranoside (IPTG), kanamycin, and Luria–Bertani (LB) medium were from Sangon (Shanghai, China).

### 2.2. Gene Cloning, Expression, and Protein Purification

Two putative GH28 family genes, *IDSPga28-4* and *IDSPga28-16*, were PCR-amplified (Appendix A) from the cDNA of Hu sheep rumen fluid using Super pfx DNA polymerase (CWBIO, Beijing, China). The expected fragments were purified and cloned into the truncated pET-30a(+) vector by *Bam*HI + *Xho*I via homologous recombination using the Trelief™ SoSoo Cloning kit (TsingKe Biotech, Beijing, China). The resulting ligation products were transformed into *E. coli* BL21 (DE3)-competent cells by heat shock transformation and streaked onto an LB agar plate (5 g/L yeast extract, 10 g/L tryptone, 10 g/L sodium chloride, 20 g/L agar) supplemented with 50 μg/mL of kanamycin, and then incubated at 37 °C for 16 h. Ten colonies of each gene construct were selected for colony-PCR validation. Positive recombinant plasmids were further confirmed by Sanger sequencing (Sangon, Shanghai, China). The recombinant hosts were designated as BL21/pET30a/*IDSPga28-4* and BL21/pET30a/*IDSPga28-16*, respectively. Subsequently, IPTG-induced gene expression, cell pellet collection, and sonication were carried out as described previously [16], and then crude enzyme, resuspended in ice-cold phosphate-buffered saline (PBS, pH 7.4), was loaded onto a HisTrap™ 5 mL column (GE Healthcare BioSciences, Pittsburgh, PA, USA) fitted to an ÄKTA start protein purification system. The bound proteins were eluted with elution buffer, supplemented with a linear gradient of 20–250 mM imidazole. Eluted fractions were analyzed by 12% sodium dodecyl sulfate-polyacrylamide gel electrophoresis (SDS-PAGE), and the purified protein fractions were used for enzymatic assays.

### 2.3. Sequence Analysis

Amino acid alignment was performed using ClustalO (https://www.ebi.ac.uk/Tools/msa/clustalo/, accessed on 10 March 2022) and BLASTp (https://blast.ncbi.nlm.nih.gov/Blast.cgi, accessed on 10 March 2022). Signal peptides were predicted using SignalP 5.0 (https://services.healthtech.dtu.dk/, accessed on 10 March 2022). The isoelectric point and molecular weight were predicted by ProtParam (https://web.expasy.org/protparam/, accessed on 10 March 2022). Multiple sequence alignment was conducted by MUSCLE with MEGA software (v.7.0, Kumar Lab, Temple University, Philadelphia, PA, USA). Phylogenetic tree analysis was performed using the Neighbor joining (NJ) statistical method based on the Poisson correction model with 1000 bootstrap replications, and then the tree was imported into the online software iTOL v6.4 (https://itol.embl.de/, accessed on 12 September 2022) for further optimization. The three-dimensional (3-D) structures were modeled by SWISS-MODEL (https://www.swissmodel.expasy.org/, accessed on 21 October 2022) and visualized using PyMOL v2.4 (https://pymol.org/2/, accessed on 21 October 2022).

### 2.4. Enzyme Activity Assay

Polygalacturonase activity was determined using the 3,5-dinitrosalicylic acid (DNS) method [21] with GalA as the standard. Briefly, purified enzyme solution (15 μL, 0.43 μg protein) and PGA (60 μL, 0.25% *w*/*v*) were dissolved in 100 mM citric acid-Na_2_HPO_4_ buffer (pH 5.0), unless otherwise stated. After incubation at 40 °C for 10 min, DNS solution (75 μL) was added to terminate the reaction, followed by 10 min in a boiling water bath, and then the absorbance of the reaction mixture was measured at 540 nm, using a Spark multimode microplate reader (Tecan, Männedorf, Switzerland). One enzyme unit (U) was defined as 1 μmol of reducing sugar released per minute. Enzyme concentration was measured by the Bradford method [22] using bovine serum albumin as the standard. To investigate the substrate specificity, purified enzyme was reacted with 0.25% (*w*/*v*) PGA, rhamnogalacturonan I, or pectin. 

### 2.5. Characterization of Purified Recombinant Enzymes

To determine the optimum pH, purified IDSPGA28-4, or IDSPGA28-16, was incubated with 0.25% (*w*/*v*) PGA over a pH range of 3.0–10.0 (pH 3.0–8.0, citric acid-Na_2_HPO_4_ buffer; pH 8.0–9.0, Tris-HCl buffer; pH 9.0–10.0, glycine-NaOH buffer) at 40 °C for 10 min, and then the reducing sugar produced was measured as described above. The optimum temperature of each enzyme was determined in buffer at the corresponding optimum pH and temperatures between 30 and 70 °C. The maximum enzyme activity was designated as 100%. All assays were performed in quadruplicate.

The pH stability was determined by measuring the residual activities of IDSPGA28-4 or IDSPGA28-16 under the optimum conditions (40 °C and pH 5.0) after preincubating the enzymes at 25 °C for 1 h in various buffers of pH 3.0–10.0. For the thermostability assay, IDSPGA28-4 or IDSPGA28-16 was pretreated at 30, 40, or 50 °C for 1 h. Aliquots were collected at different time intervals (10, 20, 30, and 60 min) and used to determine the residual activity. The initial enzyme activity before preincubation was designated as 100%. All assays were performed in quadruplicate.

### 2.6. Hydrolysis Product Analysis

Approximately 5 μg of purified enzyme was incubated with 0.25% polygalacturonic acid at pH 5.0 and 37 °C for 3 h. Aliquots sampled at different time intervals (30 min, 1 h, and 3 h) were boiled for 10 min, centrifuged at 25 °C and 10,000× *g* for 10 min, and then subjected to thin-layer chromatography (TLC) and high-performance anion exchange chromatography (HPAEC). The products formed by treating 0.25% (GalA)_n_ (n = 1, 2, 3) with the purified enzymes for 24 h were analyzed by TLC and HPAEC.

TLC was conducted using a 10 × 10 cm silica gel 60 plate (Merck, Darmstadt, Germany) with the developing solvent, 1-butanol/acetic acid/water = 2:1:1 (*v*/*v*/*v*). The plate was taken out when the solvent front was about 1 cm from the top and, after the solvent had evaporated completely, the plate was sprayed with visualization reagent (sulfuric acid: ethanol = 5:95, *v*/*v*). The plate was air-dried, then heated at 105 °C for 10 min, until spots appeared.

HPAEC was performed on an ICS-3000 system (Thermo Scientific Dionex, Sunnyvale, CA, USA) equipped with a CarboPac PA-200 column (250 mm × 2 mm; Thermo Scientific Dionex). The mobile phase (flow rate, 0.25 mL/min, at 30 °C) was composed of: (A) 0.1 M sodium hydroxide and (B) 0.1 M sodium hydroxide + 1 M sodium acetate. After sample injection, the following linear gradient was applied: 0 min, 0% B; 10 min, 10% B; 25 min, 30% B; 30 min, 100% B; 40 min, 0% B; 40–49 min 0% B.

### 2.7. Molecular Docking and Molecular Dynamics (MD) Simulation

The 3-D structures of (GalA)_2_ and (GalA)_3_ were downloaded from PubChem (https://pubchem.ncbi.nlm.nih.gov/, accessed on 1 November 2022), and the 3-D structure of IDSPGA28-16 was modeled with SWISS-MODEL (https://www.swissmodel.expasy.org/, accessed on 21 October 2022), using the known structure of *Thermotoga maritima* exo-polygalacturonase (PDB: 3JUR), which has the highest sequence similarity (41.08%) as the template. Molecular docking between IDSPGA28-16 and (GalA)_2_, or (GalA)_3_, was performed by AutoDock Vina (v1.1, CCSB, Scripps Research Institute, California, USA) [23]. A 3-D box was defined to restrict the conformational sampling space, in which the center point of the catalytic-site Asp residues (D251, D272, and D273) was set as the box center, and the box size was 30 Å, 28 Å, and 26 Å in the x-, y-, and z-dimensions, respectively. Exo-polygalacturonase hydrolyzes the first glycosidic bond from the non-reducing end of PGA, whereas exo-poly-α-digalacturonosidase hydrolyzes the second α-1,4-glycosidic bond from the non-reducing end of the substrate, according to the CAZy database. Therefore, the conformation of the protein–substrate complex, with the nonreducing end inside the catalytic pocket and the highest score from AutoDock Vina, was subjected to molecular dynamics (MD) simulation. 

The MD simulations of the complexes were carried out by Gromacs 2019.6 (https://doi.org/10.5281/zenodo.3685922, accessed on 5 November 2022) with the AMBER ff14SB force field for proteins and GAFF for the sugar molecules, using TIP3P as the water model [24,25,26]. A dodecahedral box was generated, spaced 1 nm away from the periphery of the enzyme–substrate complex, with 69,517 water molecules as the solvent to fill the box for MD simulation of the IDSPGA28-16/(GalA)_2_ complex, and 47,197 water molecules for IDSPGA28-16/(GalA)_3_. Because IDSPGA28-16 is negatively charged, 10 of the water molecules were replaced with Na^+^ ions to keep the system electrically neutral. For energy minimization, 50,000 steps of conjugate gradient minimization and a steepest descent energy minimization for every 1000 steps were conducted, with hydrogen bonds constrained by LINCS [27]. The V-rescale method was used for temperature control, the Parrinello–Rahman method for pressure coupling, and the PME method for the calculation of long-range electrostatic interactions. The van der Waals interactions were cut-off at 10 Å. The 100 ns simulation was performed in the isothermal–isobaric NPT ensemble (300 K, 1 atm) with a time step of 2 fs.

Water molecules were removed and the periodicity of the trajectory modified, using the trjconv module of Gromacs, and then the root-mean-square deviation (RMSD) for every 2 ps was calculated by the rms module. The noncovalent interactions (NCIs) of the two complexes were analyzed by Multiwfn 3.8 using the independent gradient model (IGM) and plotted by VMD v1.9.3 (http://www.ks.uiuc.edu/Research/vmd/, accessed on 14 December 2022) [28,29]. The detailed schematic diagrams of the intermolecular interactions, especially hydrogen bonds within 3.5 Å, were plotted by LigPlot^+^ (v2.2, EMBL-EBI, Wellcome Genome Campus, Cambridgeshire, UK).

### 2.8. Fruit Juice Clarification with Exo-Polygalacturonase

Fresh grape juice, orange juice, and apple juice were extracted from Kyoho grapes, Gannan Navel oranges, and Shaanxi Red Fuji apples (500 g each), respectively, then filtered through eight layers of gauze to remove pulp solids. Ascorbic acid (0.5% *w*/*v*) was added to the freshly extracted fruit juice to minimize oxidation. Purified IDSPGA28-4, or IDSPGA28-16, (~500 μg) was incubated with aliquots of juice (5 mL) at 37 °C, without stirring, for 1.5 h. Purified IDSPGA28-4, or IDSPGA28-16, was boiled for 10 min before addition to the juice, as controls. The light transmittance of the treated juice was measured with a UV/vis spectrophotometer (Phoenix, Shanghai, China) at 660 nm (%T_660_), with the light transmittance of distilled water as 100%. All reactions were performed in triplicate.

### 2.9. Statistical Analysis

All data were represented as the mean ± standard deviation (SD). Statistical analysis was performed using GraphPad Prism software (v.8.0.2, San Diego, CA, USA), via Student’s *t*-test (* *p* < 0.05; ** *p* < 0.01; *** *p* < 0.001).

## 3. Results and Discussion

### 3.1. Gene Cloning and Sequence Analysis

In the past decade, multi-omics approaches, such as metagenomics and metatranscriptomics, have been adopted widely to improve the understanding of the composition and functionality of the gastrointestinal microbiome in humans and animals [30]. There is great interest in mining novel CAZymes from the ruminant microbiome, because of their promising potential for processing of plant-derived foods and beverages. In this study, two exo-polygalacturonase genes, *IDSPga28-4* and *IDSPga28-16*, were cloned from rumen fluid cDNA. The ORFs of *IDSPga28-4* and *IDSPga28-16* encoded 541 and 458 amino acids, respectively. Multiple sequence alignment revealed that IDSPGA28-4 and IDSPGA28-16 shared the highest similarity with two GH28 proteins from an unclassified Oscillospiraceae bacterium (96.02% identity with GenBank: MBP3209358) and a Firmicutes bacterium (69.39% identity with GenBank: MBR1735702). However, neither of the latter enzymes have been functionally characterized.

Homology modeling suggested that IDSPGA28-4 and IDSPGA28-16, which have 41.08 and 37.60% identities, respectively, with the exo-polygalacturonase from *Thermotoga maritima* (PDB: 3JUR), have secondary structures mainly composed of parallel β-sheets typical of the GH28 family. In addition, each enzyme has eight conserved amino acid residues (N290, D292, D313, D314, H347, G353, R378, and K380 for IDSPGA28-4; N249, D251, D272, D273, H306, G312, R337, and K339 for IDSPGA28-16), compared with 3JUR (Appendix A) [10,31]. The overall structures revealed that a large cleft was formed by four surrounding loops in IDAPGA28-4 (T167-S202, Q225-A239, S320-N330, and N415-R437) and IDSPGA28-16 (Y97-S110, Q158-L198, S279-G289, and D373-E386) (Appendix A), which are similar to those of the exo-poly-α-digalacturonosidase, YeGH28, from *Yersinia enterocolitica* [5]. Moreover, three aspartates (D251, D272, D273) located at the bottom of the cleft have been proposed as the key catalytic residues. Phylogenetic tree analysis was performed on IDSPGA28-4, IDSPGA28-16, and those of the GH28 polygalacturonases that had been previously characterized according to their mode of action and origin (Figure 1). IDSPGA28-4 and IDSPGA28-16 were clustered on the same branch as two exo-polygalacturonases from *Bacteroides thetaiotaomicron* (GenBank: AAO79228 and AAO79260) and one from *T. maritima* (GenBank: AAD35522), indicating that these enzymes have a similar mode of action.

### 3.2. Protein Expression and Enzymatic Properties

Recombinant IDSPGA28-4 and IDSPGA28-16 were successfully expressed in *E. coli*. Two electrophoresis bands of ~70 and ~60 kDa (Figure 2) were observed, consistent with the theoretical molecular masses of the two enzymes (IDSPGA28-4 ~60 kDa; IDSPGA28-16 ~51 kDa), added to that of the linked vector-peptide, respectively. Biochemical characterization revealed that both recombinant IDSPGA28-4 and IDSPGA28-16 were optimally active at 40 °C (Figure 3A,C); the enzymes were relatively stable below 40 °C but degraded rapidly above 50 °C (Figure 3B,D). Most CAZymes isolated from ruminants have temperature optima of 30–50 °C, including those from cattle [32], buffalo [33], sheep [15,16,17,18], and goat [34]. The mesophilic adaptation of IDSPGA28-4 and IDSPGA28-16 appears to result from accommodation to the ruminant gastrointestinal temperature (38–41 °C). IDSPGA28-4 and IDSPGA28-16 were optimally catalytically active at pH 5.0 and most stable from pH 4.0 to 6.0 (Figure 3E). After pretreatment for 1 h, both enzymes were relatively stable, retaining over 60% residual activity between pH 4.0 and 9.0 (Figure 3F). 

Substrate selectivity determination indicated that both IDSPGA28-4 and IDSPGA28-16 were active against PGA, with specific activities of 31.2 ± 1.5 and 330.4 ± 12.4 U/mg (Table 1), respectively, but inactive toward pectin or rhamnogalacturonan I. Although some endo-polygalacturonases are highly active, exhibiting >20,000 U/mg against PGA [4,8], most exo-galacturonases have specific activities <300 U/mg [5,12,13,14]. Thus, IDSPGA28-16 was one of the most active exo-polygalacturonases found. The relatively high catalytic activity but poor thermostability of IDSPGA28-4 and IDSPGA28-16 suggested that protein engineering strategies such as directed evolution [35] and/or molecular cyclization [36] could improve their thermostability. 

### 3.3. Hydrolytic Products and Hydrolysis Pattern Analyses

To investigate the reaction modes of IDSPGA28-4 and IDSPGA28-16, the hydrolytic products of their action on PGA and (GalA)_n_ (n = 1, 2, 3) were determined using TLC and HPAEC. IDSPGA28-4 exclusively cleaved GalA monomers from PGA (Figure 4A,B). Both IDSPGA28-4 and IDSPGA28-16 rapidly released products from PGA, and their concentrations increased to a plateau. After reaction for 3 h, IDSPGA28-4-catalyzed hydrolysis yielded 114.5 nmol of GalA (Figure 4A,B). Exo-polygalacturonase (EC 3.2.1.67) exclusively releases GalA monomers from substrates including digalacturonate [4,8,12,37]. To further analyze the mode of action of IDSPGA28-4 and IDSPGA28-16, mono- and oligogalacturonides were used as hydrolysis substrates. As expected, IDSPGA28-4 generated only GalA from both (GalA)_2_ and (GalA)_3_ (Figure 4D,E), indicating a conventional exo-acting mode for the enzyme.

Surprisingly, IDSPGA28-16 was capable of liberating both (GalA)_2_ and GalA from PGA, although GalA was the dominant product (Figure 4B,C). After reaction for 3 h, IDSPGA28-16 generated 338 nmol GalA and 16.3 nmol (4.6% of total reducing sugars) of (GalA)_2_. In addition, IDSPGA28-16 converted (GalA)_3_ into almost equal amounts of GalA and (GalA)_2_, but did not hydrolyze (GalA)_2_ (Figure 4E,F). The co-existence of GalA and (GalA)_2_ in IDSPGA28-16 hydrolysis products suggested that the enzyme could act in a combined exo/endo-mode, i.e., IDSPGA28-16 hydrolyzed PGA into oligogalacturonides with DP 3-10, and then subsequently converted them into the monomer and dimer. 

Similar product profiles have been observed previously. Heterologous expression and characterization of five endo-polygalacturonases, BcPGs, from the plant pathogenic fungus, *Botrytis cinerea*, revealed that BcPG3 and BcPG6 mostly generated GalA and (GalA)_2_ from PGA, but trace amounts of (GalA)_3_ to (GalA)_6_ were also detected during the reaction [9]. The endopolygalacturonase PGD, derived from *A. niger*, mainly hydrolyzed PGA into GalA and (GalA)_2_ as final products, but (GalA)_3_ was also observed as an intermediate [38]. However, no intermediate oligogalacturonide products were observed during the time-course reaction of IDSPGA28-16. 

Exo-poly-α-digalacturonosidase (EC 3.2.1.82) is an exo-acting hydrolase that cleaves the second α-1,4 glycosidic bond from the non-reducing end of PGA, exclusively producing (GalA)_2_ [5,14]. However, IDSPGA28-16 produced both GalA and (GalA)_2_ as hydrolytic products, suggesting that IDSPGA28-16 had a distinct mode of action from known endo-polygalacturonases (EC 3.2.1.15), exo-polygalacturonases (EC 3.2.1.67), and exo-poly-α-digalacturonosidases (EC 3.2.1.82). The product profiles (Figure 4) suggested that both IDSPGA28-4 and IDSPGA28-16 were processive exo-polygalacturonases that performed multiple glycosidic bond cleavages during each encounter with a PGA polymer chain [9,10,37,38]. 

### 3.4. Molecular Dynamics (MD) Simulation

To understand the structural basis of substrate binding and catalysis of IDSPGA28-16, molecular docking and MD simulation analyses were performed on the enzyme structural models. As molecular docking reflects the specific state at a certain moment, the most stable conformation of the complex was selected by molecular docking and then used for MD simulation (Figure 5). MD simulation of IDSPGA28-16 with (GalA)_2_ as a substrate reached equilibrium after 30 ns, with a root-mean-square deviation (RMSD) of 5.11 Å (Figure 5A). With (GalA)_3_ as the substrate, the enzyme–substrate complex reached equilibrium in ~20 ns, with a much smaller RMSD of 1.89 Å (Figure 5B). Generally, RMSD is used as a quantitative assessment of similarity between two protein structures or protein–substrate complexes; a relatively low RMSD is preferable for mechanistic simulation [39]. In this case, the conformation with the highest score obtained by molecular docking was used as the initial conformation for MD simulation. The high RMSD of the enzyme–(GalA)_2_ complex suggested that (GalA)_2_ could bind flexibly, in more than one conformation, whereas (GalA)_3_ was relatively tightly constrained.

To further unravel the detailed molecular interactions between key active-site amino acid residues and substrates, the enzyme–(GalA)_2_ and enzyme–(GalA)_3_ representative structures obtained from MD simulations for 70–100 ns were used to visualize the non-covalent interaction (NCI) iso-surface diagram within 5 Å of the substrate (Figure 5C,D). The (GalA)_2_ or (GalA)_3_ was positioned at the bottom of the cleft formed by four surrounding loops in IDSPGA28-16 (Appendix A). It is widely accepted that eight highly conserved residues, including three aspartates, participate in substrate binding and catalysis in endo- or exo-polygalacturonases [5,10,11,30]. Seven (N249, D272, D273, H306, G312, R337, and K339) of the eight key resides were located within 5 Å of (GalA)_3_ (Figure 5). More importantly, these residues had strong binding interactions with the substrate; two catalytic aspartates, D272 and D273, formed hydrogen bonds of 2.47 Å and 3.23 Å, respectively, with the +1 subsite of (GalA)_3_ (Figure 5F). Several basic amino acids, including K278, K287, R337, and K339, also interacted with the substrates, contributing to stabilization of the enzyme–substrate complex, in good agreement with previous reports [5,10]. Mutations of the conserved basic amino acids, i.e., arginine (R256N) and lysine (K258N), in the *A. niger* endopolygalacturonase II, dramatically reduced its catalytic activity, but decreased its *K_m_* 10-fold [11]. In the MD stimulation of the enzyme–(GalA)_2_ complex, only D272 of the eight key residues was within 5 Å of the substrate (Figure 5E), consistent with the inability of the enzyme to hydrolyze (GalA)_2_ (Figure 4). 

Regarding the different acting modes of endo- and exo-polygalacturonases, the four conserved protein loops, particularly loop 1, appear to contribute strongly to substrate recognition and binding. Loop 1 in endo-polygalacturonases is involved in forming one “wall” of the “substrate path”, contributing to a tunnel-like active site, rather than orientating to the active center [4,5,10,11,40]. The substrate tends to lay along the tunnel-like active site in endo-polygalacturonase, which randomly cleaves α-1,4 glycosidic bonds via a single-attack manner, generating pectic oligogalacturonides and monomers [4,9,41,42]. In contrast, loop 1 in exo- polygalacturonase [19,30] and exo-poly-α-digalacturonosidase [5] formed the “back wall” of the substrate-binding cavity, forming to a pocket-like active site (Figure 5 and Appendix A). Interestingly, in addition to several basic amino acid residues, the acidic residue E101, positioned on loop 1, also formed three strong hydrogen bonds with the −1 subsite (Figure 5F). After PGA, or a pectic oligogalacturonide substrate was bound to the buried basic amino acid residues, its reducing end was blocked by loop 1. Consequently, exo-polygalacturonase could only access and cleave the first α-1,4 glycosidic bond from the reducing end and remove monomers progressively (Figure 4A,B) [12,19]. Unlike known exo-polygalacturonases, IDSPGA28-16 was incapable of cleaving (GalA)_2_ (Figure 4E,F), probably because of unproductive substrate binding preventing access of the catalytic residues to the α-1,4 glycosidic bond (Figure 5A,C,E). Taken together, the above stoichiometry and stereochemistry data suggested that IDSPGA28-16 was neither a typical exo-polygalacturonase nor an exo-poly-α-digalacturonosidase. It is an unconventional exo-polygalacturonase.

### 3.5. Effect of Recombinant Enzymes on Fruit Juice Clarification

Many fruits contain high concentrations of pectin, which is responsible for the high viscosity of fruit pulp and the turbidity and often poor yield of freshly pressed fruit juice. Therefore, the treatment of fruit juice, or pulp with pectinases, such as polygalacturonase, pectin lyase, and rhamnogalacturonase, to degrade pectin has been extensively used industrially to increase juice yield, reduce viscosity, and for clarification of the juice, for increased consumer acceptance [43,44]. Although IDSPGA28-4 and IDSPGA28-16 could not degrade the highly esterified (60% methylated) citrus pectin, a previous study [4] reported that exo-polygalacturonase can degrade lightly esterified pectin and clarify grape juice. The effectiveness of IDSPGA28-4 and IDSPGA28-16 for clarification of orange, grape, and apple juice was determined. Treatment with IDSPGA28-4 significantly increased the light transmittance (%T_660_) of grape juice (pH = 3.6) from 1.6% to 36.3% (*p* < 0.001) (Figure 6A), whereas IDSPGA28-16 dramatically increased the %T_660_ of apple juice (pH = 3.4) from 1.9% to 60.6% (*p* < 0.001) and grape juice from 1.1% to 43.8% (*p* < 0.001) (Figure 6B). 

The acidic pH of most fruit juices, such as orange, grape, apple, lemon, and papaya, requires pectinases with acidic pH optima for industrial application [41]. After pre-incubation at pH 3.5 for 1 h, IDSPGA28-4 and IDSPGA28-16 retained ~50% of their original activities (Figure 3E,F), displaying resilience to acidic pH comparable to many previously reported pectinases [4,45,46]. IDSPGA28-4 and IDSPGA28-16 were less effective for clarifying orange juice (pH = 5.6) than grape and apple juice, even though they had high catalytic activity from pH 5.0 to 6.0 (Figure 3E), and the effectiveness in clarifying apple juice was different between two enzymes. The reason may be that the degrees of esterification of pectin contained in these fruits are different [47] and the two enzymes differ in their preference for the degree of pectin esterification. A cocktail of endo- and exo-acting hydrolyses, as well as pectin methylesterases, could further improve the extraction efficiency and clarification of fruit juice [4,8,41], which appears to be more attributable to the variable degree of esterification in pectin.

## 4. Conclusions

In this study, two exo-polygalacturonases from sheep rumen microbiota, IDSPGA28-4 and IDSPGA28-16, were found to hydrolyze PGA, and the latter was among the most catalytically active exo-polygalacturonases to date. Distinct from IDSPGA28-4 and previously reported exo-polygalacturonases, IDSPGA28-16 was capable of liberating galacturonic acid monomers from substrates only with DP > two, suggesting that it underwent a unique action mode. IDSPGA28-4 and IDSPGA28-16 were effective for fruit juice clarification, showing potential applications in the food and feed industries.

## Figures and Tables

**Figure 1 microorganisms-11-00760-f001:**
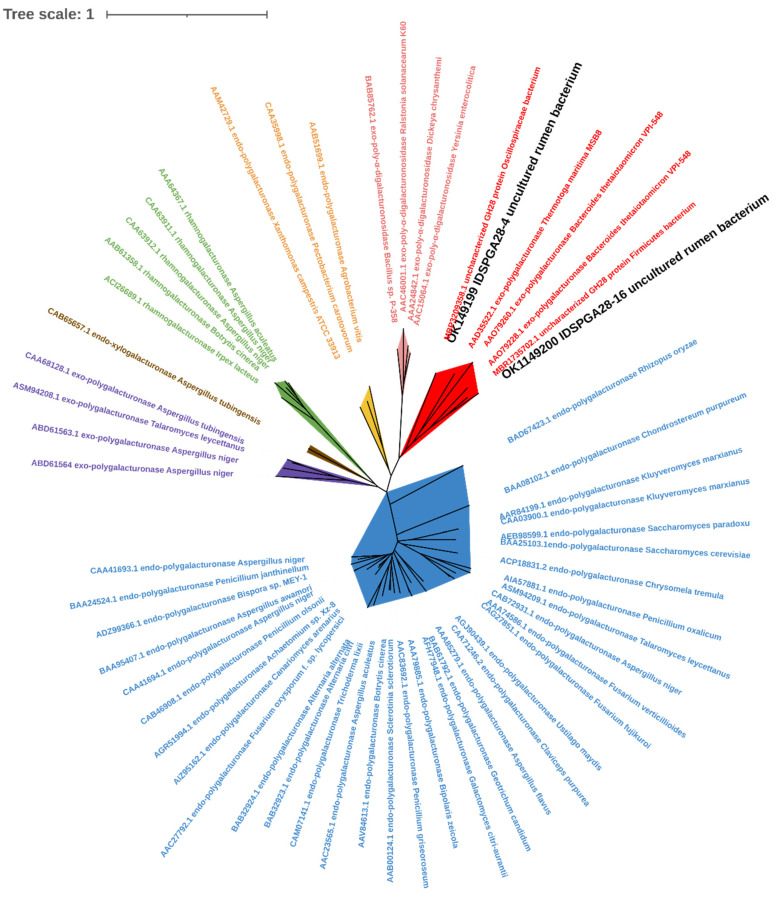
Phylogenetic analysis of IDSPGA28-4 and IDSPGA28-16. Color-coding is as follows: cherry: bacterial exo-polygalacturonase (EC 3.2.1.67); plum: bacterial exo-poly-α-digalacturonosidase (EC 3.2.1.82); yellow: bacterial endo-polygalacturonase (EC 3.2.1.15); green: fungal rhamnogalacturonase (EC 3.2.1.171); brown: fungal endo-xylogalacturonase (EC 3.2.1.-); violet: fungal exo-polygalacturonase (EC 3.2.1.67); azure: fungal endo-polygalacturonase (EC 3.2.1.15).

**Figure 2 microorganisms-11-00760-f002:**
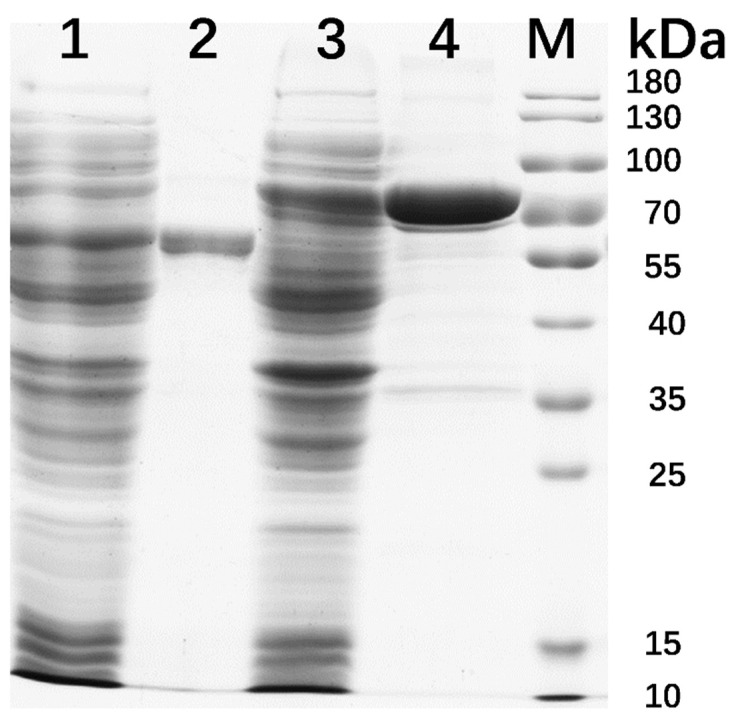
SDS-PAGE analysis of recombinant IDSPGA28-4 and IDSPGA28-16. Lane 1, crude IDSPGA28-16; 2, purified IDSPGA28-16; 3, crude IDSPGA28-4; 4, purified IDSPGA28-4; M: standard protein marker.

**Figure 3 microorganisms-11-00760-f003:**
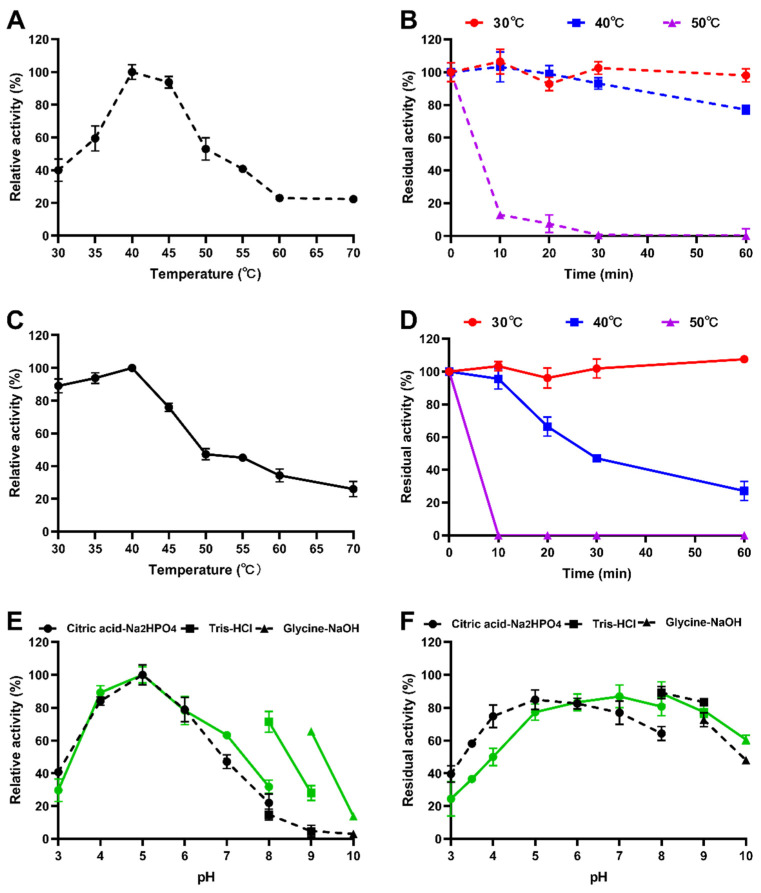
Characterization of recombinant polygalacturonases. (**A**) Optimum temperature of IDSPGA28-4; (**B**) thermostability of IDSPGA28-4; (**C**) optimum temperature of IDSPGA28-16; (**D**) Thermostability of IDSPGA28-16; (**E**) optimum pH of IDSPGA28-4 (dotted line) and IDSPGA28-16 (solid line); (**F**) pH stability of IDSPGA28-4 (dotted line) and IDSPGA28-16 (solid line). Data represent the mean ± SD (*n* = 4).

**Figure 4 microorganisms-11-00760-f004:**
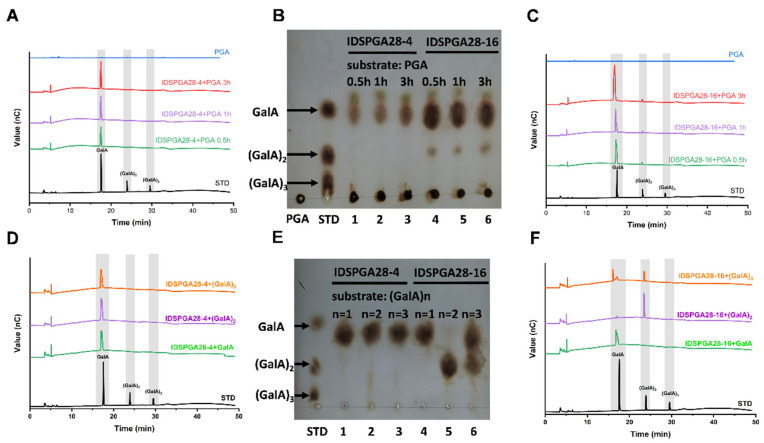
Substrates hydrolysis by IDSPGA28-4 and IDSPGA28-16, analyzed by TLC and HPAEC. (**A**–**C**) Time-course of PGA hydrolysis by IDSPGA28-4, or IDSPGA28-16. Lanes 1-3 in (**B**): IDSPGA28-4+PGA reaction for 0.5, 1, or 3 h; Lanes 4-6 in (**B**): IDSPGA28-16+PGA reaction for 0.5, 1, or 3 h; (**D**–**F**) hydrolytic pattern of pectic oligosaccharides by IDSPGA28-4 or IDSPGA28-16. Lanes 1-3 in (**E**): reaction products of IDSPGA28-4 with GalA, (GalA)_2_, or (GalA)_3_ after 24 h; lanes 4-6 in (**E**): Reaction products of IDSPGA28-16 with GalA, (GalA)_2_, or (GalA)_3_ after 24 h. PGA, polygalacturonic acid; STD, standard oligosaccharides; GalA, galacturonic acid; (GalA)_2_, digalacturonic acid; (GalA)_3_, trigalacturonic acid.

**Figure 5 microorganisms-11-00760-f005:**
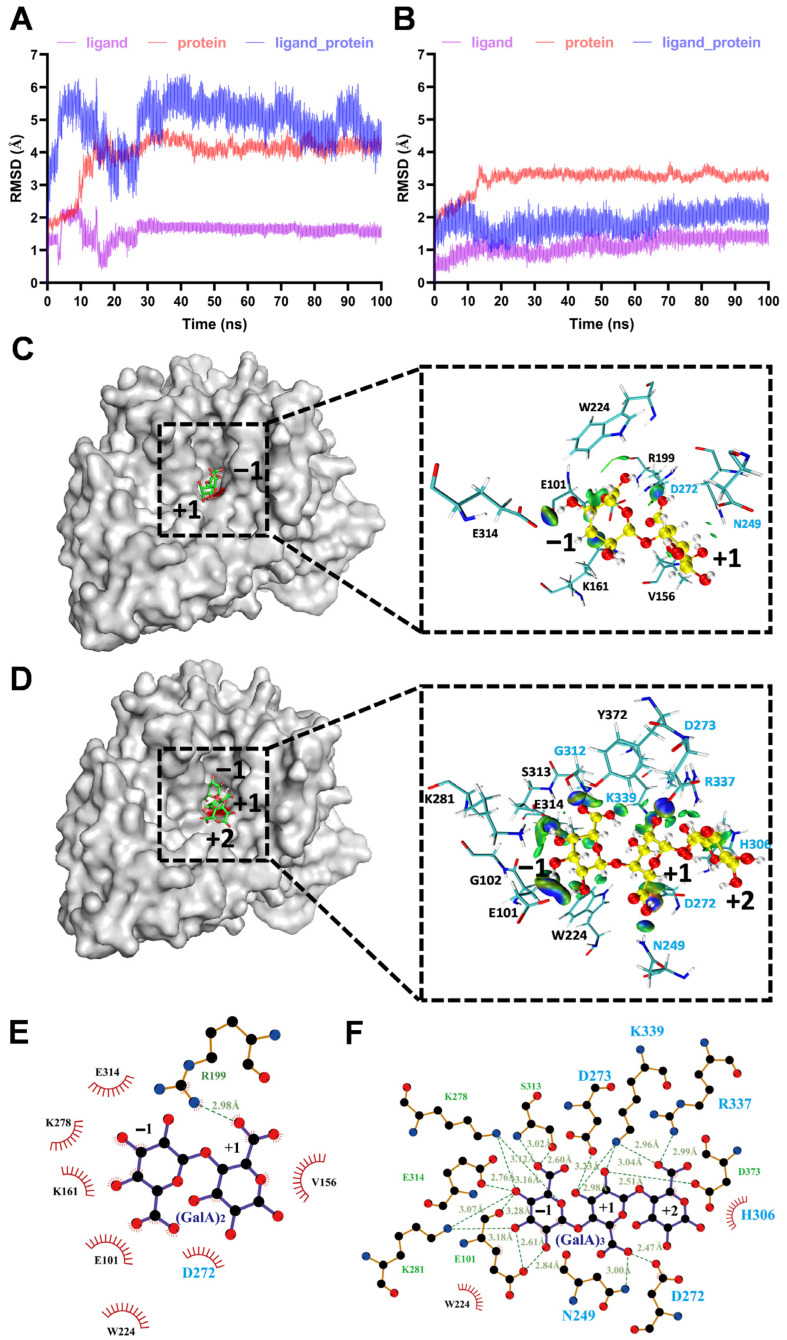
MD simulation of IDSPGA28-16 enzyme–substrate complexes. (**A**,**B**) RMSD diagram for MD simulation of IDSPGA28-16 and (GalA)_2_ (**A**) or (GalA)_3_ (**B**); (**C**,**D**) analysis of MD simulation results of IDSPGA28-16 with (GalA)_2_ (**C**) or (GalA)_3_ (**D**); representative conformation diagram of the complex clustering for 70–100 ns and the visualized NCI iso-surface diagram between IDSPGA28-16 and (GalA)_2_ or (GalA)_3_. Blue indicates hydrogen bonds and green indicates van der Waals interactions; (**E**,**F**) detailed molecular interactions between IDSPGA28-16 and (GalA)_2_ (**E**) or (GalA)_3_ (**F**).

**Figure 6 microorganisms-11-00760-f006:**
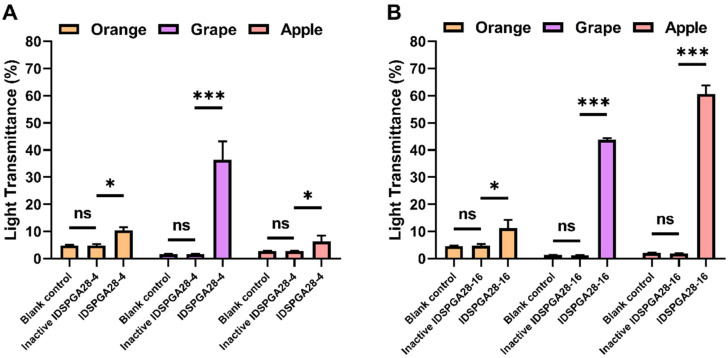
Juice clarification using IDSPGA28-4 (**A**) or IDSPGA28-16 (**B**). ns, not significant; * *p* < 0.05; *** *p* < 0.001.

**Table 1 microorganisms-11-00760-t001:** Specific activities of exo-polygalacturonases (EC 3.2.1.67) and exo-α-poly-digalacturonosidases (EC 3.2.1.82) against PGA.

Name	Origin	EC Number	Substrate (*w*/*v*)	Hydrolytic Products	Specific Activity (U/mg) ^1^	Reference
PGXA	*Aspergillus niger*	EC 3.2.1.67	0.25% PGA	GalA	7	[13]
PGXB	*Aspergillus niger*	EC 3.2.1.67	0.25% PGA	GalA	242	[13]
PGXC	*Aspergillus niger*	EC 3.2.1.67	0.25% PGA	GalA	223	[13]
Exo-PG	*Aspergillus tubingensis*	EC 3.2.1.67	0.25% PGA	GalA	230	[13]
PGAX	*Aspergillus tubingensis*	EC 3.2.1.67	0.25% PGA	GalA	255	[37]
NfPGIII	*Neosartorya fischeri*	EC 3.2.1.67	0.33% PGA	GalA	19.97	[8]
*Te*PG28a	*Talaromyces leycettanus*	EC 3.2.1.67	0.33% PGA	GalA	280 ± 9	[4]
IDSPGA28-4	rumen bacterium	EC 3.2.1.67	0.25% PGA	GalA	31.2 ± 1.5	this study
IDSPGA28-16	rumen bacterium	EC 3.2.1.67	0.25% PGA	GalA, (GalA)_2_	330.4 ± 12.4	this study
*Ye*GH28	*Yersinia enterocolitica*	EC 3.2.1.82	1% PGA	(GalA)_2_	80	[5]
PehK	*Bacillus subtilis*	EC 3.2.1.82	0.75% PGA	(GalA)_2_	33.1	[14]

^1^ Data represent the mean or mean ± SD (*n* = 3 or 4).

## Data Availability

The datasets generated during and/or analyzed during the current study are available from the corresponding author on reasonable request.

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
