# Peer review of "Characterization of Two Novel Rumen-Derived Exo-Polygalacturonases: Catalysis and Molecular Simulations"

_microorganisms, 2023, doi:10.3390/microorganisms11030760_

Round 1

Reviewer 1 Report

I read carefully the paper “Characterization of two novel rumen-derived exo-2 polygalacturonases: catalysis and molecular simulations” and I found it quite interesting. The review has a certain value, but there are still many aspects to be improved and supplemented.

Abstract:

L 18-20: rephase “novel pectinases .... novel pectinases”

l27:Before indicate wich substrates were tested

Introduction

The introduction provides sufficient background to the study, all the cited references are relevant to the research and the research design is appropriate. However, I suggest to strengthen the depth and breadth of the introduction and summarize the research status, rather than simply listing the literature.

The authors did not define the objective and hypothesis of your research.

L40-42: Please support it with a reference.

Materials and methods

Methods are adequately described, but the number of replicates should be explained better in each subsection.

L105: Please indicate the conditions and time of storing.

L287-288:Did you check possible outliers?

L288-289: Please explain the tested groups or treatments. If did you test 3 juice, why used a one via test?

L188-190. This is not a conclusion

The format of references is not uniform, please revise carefully. For example, the following document "Monhnen: D. Pectin structure and ..."

Author Response

I read carefully the paper “Characterization of two novel rumen-derived exo-2 polygalacturonases: catalysis and molecular simulations” and I found it quite interesting. The review has a certain value, but there are still many aspects to be improved and supplemented.

Abstract:

L 18-20: rephase “novel pectinases .... novel pectinases”

Response: We have rephrased the sentence. Please refer to lines18-20.

l27: Before indicate which substrates were tested

Response: We indicated the substrate tested as polygalacturonic acid. Please refer to line 27.

Introduction

The introduction provides sufficient background to the study, all the cited references are relevant to the research and the research design is appropriate. However, I suggest to strengthen the depth and breadth of the introduction and summarize the research status, rather than simply listing the literature.

The authors did not define the objective and hypothesis of your research.

Response: Thanks for your kind advice. We modified the relevant content accordingly in Introduction. Please refer to Lines 95-98 in revised manuscript.

L40-42: Please support it with a reference.

Response: We provided the relevant references in revised manuscript.

  1. Caffall, K.H.; Mohnen, D. The structure, function, and biosynthesis of plant cell wall pectic polysaccharides. Carbohyd Res 2009, 344, 1879–1900.
  2. Atmodjo, M.A.; Hao, Z.; Mohnen, D. Evolving views of pectin biosynthesis. Annu. Rev. Plant Biol. 2013, 64, 747–779.

Materials and methods

Methods are adequately described, but the number of replicates should be explained better in each subsection.

Response: Thanks for suggestion. We have indicated the number of replicates as “All assays were performed in quadruplicate’. Please kindly refer to lines 186-187 and 196 in revised manuscript.

L105: Please indicate the conditions and time of storing.

Response: We indicated the condition and time of storing as suggested.

L287-288: Did you check possible outliers?

Response: Yes, we had checked possible outliers via Grubbs test by using GraphPad Prism software (v.8.0.2, San Diego, CA, USA), and no outlier was found

L288-289: Please explain the tested groups or treatments. If did you test 3 juice, why used a one via test?

Response: We conducted clarification of three fruit juice, including orange juice, grape juice and apple juice, by using IDSPGA28-4 and IDSPGA28-16 in this study. Let’s take effects of IDSPGA28-4 on orange juice for example (left three columns in Figure6, panel A). The first column is light transmittance of untreated orange juice (Blank control). The second column is light transmittance of orange juice treated by inactive IDSPGA28-4 (heat-treated enzyme, Negative control). The third column is light transmittance of orange juice treated by active IDSPGA28-4 (Test group. The results show that the light transmittance of orange juice treated by active IDSPGA28-4 is significantly higher than that by inactive enzyme (P<0.05), suggesting that active IDSPGA28-4 treatment contributes to juice clarification probably by degrading the pectin in orange. However, there is no difference between Blank control and Negative control. Meanwhile, we tested the effects of IDSPGA28-4 on grape and apple juice.

L188-190. This is not a conclusion

Response: We have revised it accordingly. Please refer to lines 188-190 in Conclusion section.

The format of references is not uniform, please revise carefully. For example, the following document "Monhnen: D. Pectin structure and ..."

Response: We have corrected it accordingly and checked all references cited as well.

Dear authors, the manuscript entitled" Characterization of two novel rumen-derived exo-polygalacturonases: catalysis and molecular simulations" has a very interesting topic.

The gastrointestinal microbiome in humans and animals is the key to a healthy individuals. The  manuscript is a well written one. I have some suggestions:

Please describe few data related to the gut microbiota. And also describe the criteria based on witch you choose to investigate only some types of the bacteria from the gastrointestinal microbiota.

Response: Thanks for kind suggestion. Actually, the data regarding the metatranscriptomics of the Hu sheep rumen microbiome was announced and published by our group previously (He et al., 2019). Due to the extreme growth conditions of rumen in ruminants, pure culture of most microbes in rumen remains challenging. Based on sequence and catalytic domain analyses, 173 ORFs were predicted as GH28 family proteins which potentially act as pectinase. Then, we randomly picked ten of these 173 ORFs for gene cloning, and obtained two full-length ORFs, IDSGH28-4 and IDSGH28-16. In this study, we mainly described the expression in E. coli and characterized biochemical properties of the recombinant enzymes. According to amino acid sequence alignment, the IDSGH28-4 and IDSGH28-16 were close to Oscillospiraceae bacterium (96.02% identity with GenBank: MBP3209358) and Firmicutes bacterium (69.39% identity with GenBank: MBR1735702), respectively. Please refer to Lines 95-98 and 307-309 in revised manuscript.

For your reference:

  1. He, B.; Jin, S.W; Cao, J.W; Mi, L.; Wang, J.K. Metatranscriptomics of the Hu sheep rumen microbiome reveals novel cellulases. Biotechnol. Biofuels 2019, 12, 153.

Reviewer 2 Report

Dear authors, the manuscript entitled" Characterization of two novel rumen-derived exo-polygalacturonases: catalysis and molecular simulations" has a very interesting topic.

The gastrointestinal microbiome in humans and animals is the key to a healthy individuals. The  manuscript is a well written one. I have some suggestions:

1. Please describe few data related to the gut microbiota. And also describe the criteria based on witch you choose to investigate only some types of the bacteria from the gastrointestinal microbiota.

Thank you!

Regards!

Author Response

(The authors gave the same response as above.)
